# Management of Anorexia–Cachexia Syndrome in a Community Palliative Care Support Team

**DOI:** 10.3390/jcm14176167

**Published:** 2025-08-31

**Authors:** Inês Saura, Joana Brandão Silva, Daniela Cunha, Iliana Ramos, Valéria Semedo, José Paulo Andrade, Marília Dourado, Hugo Ribeiro

**Affiliations:** 1Faculty of Medicine, University of Coimbra, 3000-548 Coimbra, Portugal; ines.carlota.saura@gmail.com (I.S.); jrbesilva@gmail.com (J.B.S.); mdourado@fmed.uc.pt (M.D.); 2Health Family Unit Aldeias do Xisto, Local Health Unit of Coimbra, 3004-561 Coimbra, Portugal; 3Abel Salazar Institute of Biomedical Sciences, 4050-313 Porto, Portugal; 4Health Family Unit Marco, Local Health Unit Tâmega-Sousa, 4564-007 Penafiel, Portugal; 5Northern School of Health of the Portuguese Red Cross, 3720-126 Aveiro, Portugal; daniela.fa.cunha@gmail.com; 6Department of Biomedicine, Faculty of Medicine of the University of Porto, 4200-319 Porto, Portugal; iliana_ramos@hotmail.com (I.R.); semedovaleria@gmail.com (V.S.); jandrade@med.up.pt (J.P.A.); 7Community Palliative Care Team Gaia, Local Health Unit Gaia and Espinho, 4434-502 Vila Nova de Gaia, Portugal; 8Palliative Care Unit, University Hospital Agostinho Neto, Cidade da Praia 7600, Cape Verde; 9Faculty of Science and Technology, University of Cape Verde, Praia 7943, Cape Verde; 10RISE-Health, Faculty of Medicine, University of Porto, 4200-319 Porto, Portugal; 11Coimbra Institute for Clinical and Biomedical Research, 3000-548 Coimbra, Portugal

**Keywords:** anorexia–cachexia syndrome, palliative care, home-based care, dysphagia, nutritional status, fatigue, xerostomia, enteral nutrition, functional assessment, symptom management

## Abstract

**Background/Objectives:** Anorexia–Cachexia Syndrome (ACS) is a multifactorial condition common in advanced chronic illnesses, leading to significant impacts on prognosis and quality of life. This retrospective cohort study aimed to evaluate the prevalence, management strategies, and clinical and patient-centered outcomes of ACS in a home-based palliative care team. **Methods:** Clinical records of 128 adult patients followed between 2021 and 2024 were analyzed. Data collected included sociodemographic variables, clinical diagnosis, nutritional parameters (Palliative Performance Scale (PPS), Mini Nutritional Assessment (MNA)), symptoms (anorexia, fatigue), interventions (enteral nutrition, psychological and rehabilitative support), and relevant medications. Statistical analysis included descriptive, inferential, and multivariable proportional hazard regression analysis to identify independent predictors of weight loss and anorexia. **Results:** Manifestations of ACS were observed across both oncologic and non-oncologic conditions. The prevalence of weight loss and anorexia were interrelated and were not different between diagnostic groups. Using multivariable analysis, higher baseline MNA scores (HR = 3.797, *p* = 0.006) and the use of enteral nutrition (HR = 7.418, *p* = 0.014) were independently associated with an increased risk of significant weight loss. Lower baseline PPS scores (HR = 0.069), use of enteral nutrition (HR = −0.890), and the presence of psychological support were protective for subsequent anorexia. Dexamethasone use was associated with greater nutritional decline in univariate models. **Conclusions:** The management of ACS in home palliative care requires the early identification of symptoms, multidisciplinary intervention, and personalized strategies beyond disease etiology. Risk of weight loss is associated with higher MNA scores, and these are best managed in the first week. In anorexia cases, psychological support is protective.

## 1. Introduction and Objectives

Anorexia–Cachexia Syndrome (ACS) is a highly prevalent and severe clinical entity in patients with chronic diseases in terminal stages, constituting a multidimensional phenomenon that transcends mere weight loss. In palliative care, ACS is particularly relevant, affecting approximately 80% of cancer patients in advanced stages of the disease [1,2]. This syndrome is characterized by the involuntary loss of lean and adipose body mass, anorexia, and profound metabolic changes, resulting in an adverse functional and prognostic impact, as well as a significant worsening of the psychological burden on the patient and their caregivers [2,3,4].

ACS is a state of systemic hypercatabolism, driven by inflammatory cytokines, neuroendocrine alterations, and metabolic dysfunctions, with systemic manifestations that go beyond the scope of nutrition [5,6].

The consensual definition proposed by Fearon et al. (2011) [7] establishes that the diagnosis of ACS implies the existence of a weight loss greater than 5% in six months or greater than 2% when associated with a body mass index (BMI) less than 20 kg/m^2^ or even the concomitant presence of sarcopenia. This classification proposal has allowed the stratification of patients into precachexia, cachexia, and refractory cachexia, promoting a more targeted approach to the management of the syndrome [2,7].

In palliative care, Anorexia–Cachexia Syndrome (ACS) is a particularly relevant concern, affecting a significant proportion of patients with advanced chronic illnesses [1,2]. While the impact on overall survival in this population may be complex, ACS profoundly affects quality of life. The syndrome’s characteristics—involuntary weight loss, anorexia, and metabolic changes—lead to significant functional decline, an increased symptom burden (e.g., fatigue, weakness), and a diminished ability to participate in daily activities and social interactions. These factors contribute to a substantial worsening of the psychological well-being of both patients and their caregivers [2,3,4,5,6,7,8]. The perpetuation of a cycle of fatigue, physical inactivity, and global deterioration contributes to the loss of autonomy and an increase in the patient’s burden of suffering [1,3]. In parallel, anorexia, often experienced as a rupture in the shared experience of food in a family context, represents an additional source of emotional suffering [3,9].

Difficulties in accessing specialized support, sociocultural heterogeneity, and economic barriers impact the effectiveness of therapeutic interventions [9]. The need to adapt strategies to the home reality, to train caregivers, and to establish realistic therapeutic expectations becomes imperative to ensure the quality and humanization of care [10]. Research on ACS has largely focused on hospital-based oncology populations, leaving a gap in our understanding of its dynamics in home palliative care, where we see some specific characteristics, such as resource constraints, caregiver involvement, the home environment itself, continuity of care, and patient autonomy. These operational differences highlight the need for research specifically focused on ACS management within community palliative care. Understanding the unique challenges and opportunities of this setting is essential to develop effective, patient-centered interventions that optimize quality of life for individuals with advanced illness.

Experiences in other clinical areas, such as pain management, offer valuable insights. For example, integrated scales have improved therapeutic decisions in pain management [11]. A similar, structured approach to ACS—incorporating standardized assessments of nutritional status, symptom burden, and functional decline—could optimize management, particularly in addressing the multiple and interconnected symptoms characteristic of this syndrome. Furthermore, “minor” or often-underestimated symptoms, like xerostomia and dysgeusia, can significantly impact the clinical trajectory of ACS. While sometimes overlooked, these symptoms can severely affect nutritional intake and quality of life, leading to poorer outcomes if left unaddressed [12,13]. We hypothesize that proactively managing these seemingly minor aspects can significantly influence the overall course of ACS in community palliative care patients, and this is an essential feature of the current analysis.

This retrospective study aims to analyze the prevalence, management strategies, and outcomes (patient-centered and clinical-centered outcomes) of ACS in the context of a community palliative care support team. By evaluating patient records, we aim to analyze the patterns of nutritional decline, symptom burden, and treatment approaches related to Anorexia–Cachexia Syndrome (ACS) in patients receiving home palliative care, evaluate the effectiveness of treatments, and reflect on potential improvements in daily practices. The primary objectives are as follows: (a) to determine the prevalence of ACS in patients followed by a community palliative care support team; (b) to evaluate the effectiveness of the clinical approach and management of ACS by a community palliative care support team. The secondary objectives are as follows: (a) to identify risk factors and comorbidities associated with the development of ACS in those patients; (b) to evaluate the impact of ACS on the clinical evolution and quality of life of patients; (c) to provide recommendations to improve the management of ACS in community palliative care.

## 2. Material and Methods

Study design: An observational, retrospective cohort study was carried out to analyze the prevalence, management strategies, and clinical and patient-centered outcomes, in the context of ACS, in a community palliative care support team. Population and sample: Participants were selected from the team’s databases, corresponding to patients followed between 2021 and 2024. Adult patients followed by the team and with complete clinical records were included in the study. Patients with insufficient clinical data (*n* = 365) and patients followed by the team for periods of less than 30 days (*n* = 778) were excluded. The final sample consisted of 128 patients (Figure 1).

Data collection procedures: Data collection was performed by reviewing electronic clinical records. Data were extracted regarding sociodemographic characteristics (sex, age at admission and age at death), primary diagnosis (oncological disease, dementia, heart failure, frailty syndrome), weight and body mass index (BMI) at the time of admission and at the end of follow-up, presence of dysphagia (yes or no), taste alterations (yes or no) and xerostomia (yes or no), nutritional support interventions (placement of nasogastric tube or gastrostomy), support from psychology and rehabilitation nursing, functional and nutritional assessment (Palliative Performance Scale—PPS [14]; Mini Nutritional Assessment—MNA [15]) at the beginning and at the end of the first week. Functional status was assessed using the Palliative Performance Scale (PPS) [14], a widely used tool in palliative care that assesses a patient’s ability to ambulate and perform activities of daily living, level of consciousness, and nutritional intake. The PPS ranges from 0% (bedridden, total care required) to 100% (fully functional and independent). PPS scores were determined by nurses and medical doctors based on their clinical assessment of the patient.

Symptoms of anorexia and fatigue were extracted at the time of joining the team and at the end of the first week, and medications prescribed to control nausea and vomiting and promote improved appetite. Anorexia and fatigue were assessed using a numerical rating scale ranging from 0 to 10, where 0 represents “no fatigue” and 10 represents “worst possible fatigue”. Patients were asked to rate their current level of fatigue at the time of each assessment. A score of 4 or higher was considered to indicate clinically significant fatigue. Anorexia and fatigue were assessed by nurses at the time of the initial assessment and at the end of the first week of follow-up. Whenever we could not rely on self-reporting, we applied a peer-review scale and consolidated results into three groups (mild, moderate, or severe), corresponding to values of 1–3, 4–7, and 8–10. Using a weighted average, to include all data, we considered 2, 5.5, and 9 as numerical reference values. Statistical analysis did not reveal significant confounding in converting descriptive categories to numerical ones.

Decisions regarding artificial nutrition were made on a case-by-case basis through a multidisciplinary team assessment that considered the patient’s overall disease trajectory, global functionality (as measured by PPS), estimated survival expectancy, and the degree and quickness of decline in oral intake. In accordance with Magalhães et al. (2025) [16], percutaneous endoscopic gastrostomy (PEG) was considered for patients with a survival expectancy of more than 4 weeks and an anticipated prolonged inability to maintain adequate oral intake. For patients experiencing a sudden loss of oral intake requiring immediate nutritional support, a nasogastric tube (NGT) was placed as a temporary measure while awaiting PEG placement. PEG placement typically occurred within 1–2 weeks, taking into consideration the patient’s expected survival benefit and the balance of risks and benefits. All decisions were made in consultation with the patient (if able) and their family, respecting their values and preferences.

For patients who did not receive enteral nutrition, management focused on optimizing oral intake through a combination of strategies. Patients were offered a “free diet” of preferred foods, as tolerated. Dietary modifications, such as pureed foods or thickened liquids, were used to address dysphagia. Oral nutritional supplements were provided to augment caloric intake. In some cases, the focus shifted to “comfort feeding,” prioritizing the patient’s enjoyment of food, even if nutritional goals were not fully achieved. Symptom management, including interventions for nausea, pain, and xerostomia, was a key component of care.

Ethical considerations: This study was conducted in accordance with the ethical principles set out in the World Medical Association’s Declaration of Helsinki (2013 revision) [17], as well as with European legislation on the protection of personal data, in particular Regulation (EU) 2016/679 of the European Parliament and of the Council of 27 April 2016 (General Data Protection Regulation—GDPR) [18]. The research was reviewed and approved by the Ethics Committee of the Faculty of Medicine of the University of Coimbra (FMUC), under number CE-172/2024, issued following a meeting held on 20 November 2024. It was also approved by the Ethics Committee of the Local Health Unit Gaia and Espinho (ULS GE), under number CE/2023/34.

Statistical analysis: The statistical analysis involved descriptive statistics (absolute and relative frequencies, means, and respective standard deviations) and inferential statistics. The selection of statistical tests was guided by the nature of the variables and the specific research questions being addressed. The following tests were used: the Pearson correlation coefficient (assessing linear relationships between continuous variables), the Student’s *t*-test for independent samples (comparing means of two groups), the One-Way Anova test (comparing means of multiple groups), the Kruskal–Wallis test (comparing distributions of multiple groups when data is not normally distributed), the Mann–Whitney test (comparing distributions of two groups when data is not normally distributed), the chi-square test of independence (analyzing associations between categorical variables), and the Fisher test (analyzing associations between categorical variables when expected cell counts are small). The chi-square assumption that no more than 20% of cells have expected frequencies lower than 5 was analyzed. In situations where this assumption was not satisfied, the chi-square test by Monte Carlo simulation was used. The differences were analyzed with the support of the standardized adjusted residuals. The normality of distribution was analyzed with the Shapiro–Wilk test and the homogeneity of variances with the Levene test. The significance level to reject the null hypothesis was set at α ≤ 0.05.

Multivariable logistic regression analysis was performed to identify independent clinical and demographic factors associated with weight loss and anorexia, considering the other variables. The predictive model was created using logistic and linear regression analysis and dimension reduction through the backward feature elimination method. Variables with clinical relevance included in the multivariable analysis were all found to be associated with the outcomes in the univariable analysis at a significance level of *p* < 0.25. To ensure that potentially relevant clinical factors were not excluded from the multivariable analysis, a less stringent significance level of *p* < 0.25 was used as a threshold for variables’ inclusion in the initial model. This approach is recommended in situations where there is a desire to explore a wider range of potential predictors.

The statistical analysis was performed using SPSS (Statistical Package for the Social Sciences) version 28 for Windows [19].

## 3. Results

### 3.1. Sociodemographic and Clinical Characteristics

This cohort consisted of a total of 128 patients followed by a community palliative care team, in the context of home intervention. The mean age at admission was 82.7 years (±14.4), while the mean age at death was 82.5 years (±14.4). There was an equal distribution between the sexes, with a slight predominance of females (52.3%; *n* = 67) in relation to males (47.7%; *n* = 61). The most frequent underlying pathologies in the study population were oncological pathology (47.7%; *n* = 61), frailty syndrome (23.5%; *n* = 30), dementia (14.8%; *n* = 19), and heart failure (14.1%; *n* = 18). This distribution reveals a clear predominance of cancer patients but also a substantial representation of clinical situations of high functional complexity, such as frailty and dementia. Notably, chronic kidney disease (CKD) was not represented in this cohort due to exclusion criteria related to data completeness. Table 1 shows the main sociodemographic characteristics.

The mean age at admission varied across diagnostic groups but without statistical significance (*p* > 0.05). Patients with dementia were, on average, slightly older (85.47 ± 11.51 years) compared to patients with cancer (83.38 ± 10.74 years), frailty syndrome (83.00 ± 9.48 years), and heart failure (84.67 ± 8.76 years).

The differences in the number of days of follow-up according to the type of pathology were not statistically significant (χ^2^ (3) = 1.604, *p* = 0.658).

### 3.2. Prevalence of Anorexia–Cachexia Syndrome

All patients (100%) included in this study were diagnosed with Anorexia–Cachexia Syndrome.

### 3.3. Multivariate Analysis of Factors Associated with Weight Loss and Anorexia

Multivariable logistic regression analysis was performed to identify independent factors associated with weight loss and anorexia, adjusting for potential confounders. Table 2 presents the results of the multivariable model for weight loss, and Table 3 presents the results for anorexia.

As shown in Table 1, higher initial MNA scores (HR = 3.797, 95% CI: 1.473–9.788, *p* = 0.006) and the use of NGT or PEG (HR = 7.418, 95% CI: 1.506–36.525, *p* = 0.014) were independently associated with an increased risk of weight loss > 5%. Furthermore, those who took medicine at the first week exhibited a protective hazard against weight loss > 5% (HR = 0.282; 95% CI: 0.120–0.660; *p* = 0.004). Gender was not significantly associated with weight loss in this model (*p* = 0.53).

The risk of “Anorexia After” was independently associated with a lower baseline PPS (HR = 0.069, 95% CI: 0.031–0.107, *p* < 0.001) and use of NGT or PEG (HR = −0.890, 95% CI = −1.555–−0.224, *p* = 0.009), while psychological support was a protective factor (HR = −1.578; *p* = 0.04), and those who took medicine at the first week exhibited a protective hazard against anorexia after (HR= −0.404, *p* < 0.001).

### 3.4. Univariate Analyses

The statistical analysis of final fatigue revealed significant differences according to the underlying pathology (F(3, 117) = 3.652; *p* = 0.015). Patients with an oncological diagnosis presented higher mean final fatigue values (M = 6.0 ± 1.9) compared to patients with dementia (M = 4.3 ± 2.1), suggesting a greater symptom burden and the need for differentiated symptomatic control in this group.

The correlation coefficients between initial anorexia and initial fatigue values were significant, positive, and moderate (r = 0.459, *p* = 0.001). The correlation coefficients between initial and final anorexia and final fatigue values were significant, positive, and moderate (r = 0.180, *p* = 0.049; r = 0.576, *p* = 0.001). Thus, the higher the anorexia values, the higher the fatigue values.

The correlation coefficients between age at admission, age at death, and initial and final fatigue were not statistically significant (*p* > 0.05). The differences in initial fatigue according to the type of pathology were not statistically significant (χ^2^ (3) = 3.194, *p* = 0.363).

A correlation analysis revealed a statistically significant inverse association between admission anorexia levels and BMI at the end of follow-up (r = −0.421; *p* = 0.023).

There were no statistically significant correlations between age at admission or age at death and values of anorexia (initial or final), weight, BMI, weight loss, or BMI loss (all *p* > 0.05). Likewise, the differences in values of anorexia (initial and final), weight, BMI, weight loss, and BMI loss according to the type of pathology did not reach statistical significance (all *p* > 0.05). There were also no significant correlations between anthropometric parameters (initial and final weight and BMI) and fatigue levels.

In the subgroup of individuals with dysphagia, it was found that the initial weight was significantly higher (M = 65.31 ± 12.24) than in the group without dysphagia (M = 59.62 ± 10.45; t(79) = 2.257; *p* = 0.027). However, the cumulative weight loss was more significant among patients with dysphagia (9.95% vs. 4.43%; *p* = 0.002). This group also showed higher values of final fatigue (M = 5.90 ± 2.42 vs. M = 5.31 ± 1.62; *p* = 0.042) and final anorexia (M = 4.54 ± 3.23 vs. M = 3.08 ± 2.07; *p* = 0.047).

The use of a nasogastric tube (NGT) or percutaneous endoscopic gastrostomy (PEG) was associated with a more marked weight loss (M = 11.28 ± 5.94) compared to patients without NGT/PEG (M = 5.62 ± *p* = 0.005). Interestingly, final fatigue levels were lower in patients with this type of nutritional support (M = 4.90 ± 2.36 vs. M = 5.95 ± 2.36; *p* = 0.038).

Individuals with taste alterations presented greater weight loss (M = 10.27 ± 5.71 vs. M = 4.98 ± 5.70; *p* = 0.002) and higher levels of final fatigue (M = 6.24 ± 1.71 vs. M = 5.31 ± 2.06; *p* = 0.030).

In the case of xerostomia, statistically significant associations were identified with weight loss (M = 9.66 ± 6.77 vs. M = 5.02 ± 5.23; *p* = 0.012), final fatigue (M = 6.08 ± 2.13 vs. M = 5.17 ± 1.83; *p* = 0.009), and final anorexia (M = 4.52 ± 3.18 vs. M = 3.06 ± 2.08; *p* = 0.034).

Table 4 shows the nutritional and symptomatic parameters according to the presence of dysphagia, use of NGT/PEG, taste alterations, and xerostomia.

A statistically significant association was found between the presence of taste alterations and the occurrence of xerostomia, with 75.8% of patients with xerostomia presenting taste alterations, compared to only 31.9% among those without xerostomia (Fisher’s test, *p* < 0.001). Additionally, a significant association was also observed between dysphagia and xerostomia: the proportion of patients with dysphagia was higher in the group with xerostomia (57.4%) than in the group without xerostomia (31.4%), with *p* = 0.003.

Regarding the Palliative Performance Scale (PPS), the initial values showed negative correlations with the final weight (r = −0.281; *p* = 0.025), initial fatigue (r = −0.201; *p* = 0.021), and initial anorexia (r = −0.285; *p* = 0.005). The PPS at the first week maintained this pattern, correlating negatively with the initial weight (r = −0.269; *p* = 0.011), final weight (r = −0.262; *p* = 0.014), initial fatigue (r = −0.312; *p* = 0.002), and final fatigue (r = −0.252; *p* = 0.008).

The variations in PPS according to the pathology were also statistically significant (F(3, 121) = 4.240; *p* = 0.007), with the greatest reduction being seen in cancer patients (M = −16.66 ± 10.36), as opposed to those with dementia (M = −7.36 ± 7.33), which may reflect the more abrupt and symptomatic clinical course of terminal cancer decline.

It was found that the initial PPS values were significantly higher in patients with taste alterations (29.39 vs. 25.16; Mann–Whitney U test = 1117.000; *p* = 0.023). Similarly, patients with xerostomia presented higher PPS values both at the initial evaluation (30.56 vs. 23.00; MU = 1283.500; *p* < 0.001) and after one week (13.89 vs. 11.30; MU = 1563.500; *p* = 0.020).

Table 5 shows the initial PPS values and those after one week correlated with anthropometric parameters and symptoms such as fatigue, anorexia, taste and xerostomia.

The differences in PPS according to NGT/PEG were not significant (*p* > 0.05).

The administration of dexamethasone was associated with a significantly greater reduction in PPS values (M = −16.98 ± 9.27) compared with patients not medicated with this drug (M = −10.48 ± 11.22), a statistically significant difference (*p* = 0.001).

On the other hand, patients treated with mirtazapine showed greater reductions in PPS, although this difference was not statistically significant (*p* = 0.252). Conversely, patients treated with ondansetron showed less pronounced reductions in PPS, although this difference did not reach statistical significance (*p* = 0.992).

There was a positive, moderate, and statistically significant correlation between the initial MNA and the MNA values at week one (r = 0.694; *p* < 0.001). The initial MNA showed significant positive correlations with initial weight, final weight, initial BMI, and final BMI, as well as significant negative correlations with initial fatigue levels and both initial and final anorexia values. Similarly, the final MNA correlated positively with final weight, initial, and final BMI and negatively with weight loss, fatigue (initial and final), and anorexia (initial and final).

Table 6 shows the Pearson correlations between the MNA values, both at admission and at the end of the first week of follow-up, and anthropometric parameters (weight and BMI), as well as some relevant symptoms, such as fatigue and anorexia.

The differences in initial MNA values according to diagnosis were statistically significant, with cancer patients presenting significantly higher values when compared to patients with frailty syndrome (F(3, 117) = 3.172; *p* = 0.027). Similarly, the MNA after one week of follow-up varied significantly according to the diagnosis (F(3, 117) = 3.036; *p* = 0.032), with patients with dementia standing out as presenting higher values than those observed in patients with frailty syndrome.

Table 7 presents the mean MNA values at admission and after the first week of follow-up, categorized by the primary diagnosis. A variation in nutritional status can be observed according to the underlying pathology, with a particular emphasis on cancer patients, who present a greater decline in MNA over the first week.

The differences in MNA values’ variation between the initial moment and the first week were statistically significant, according to the diagnosis (F(3, 117) = 7.128; *p* < 0.001). Patients with an oncological pathology presented a significantly more pronounced reduction in MNA (M = −5.55 ± 3.45) compared with patients with dementia (M = −2.68 ± 2.64) or heart failure (M = −2.52 ± 2.83), indicating a greater impairment of nutritional status in this clinical group.

There were no statistically significant correlations between age (at admission or date of death) and MNA values, nor between age and the variation in these values over the first week (all with *p* > 0.05). Similarly, the differences in MNA values according to sex did not reach statistical significance. Although male patients showed, on average, more pronounced reductions in MNA, this difference was also not statistically significant (*p* > 0.05).

Patients with dysphagia presented significantly higher initial MNA values (t(119) = 2.804; *p* = 0.006), while patients with taste alterations showed significantly lower final MNA values (t(119) = 2.745; *p* = 0.007). Similarly, patients with xerostomia presented significantly lower final MNA values (t(119) = 4.501; *p* < 0.001). On the other hand, MNA values (initial and final) did not present statistically significant differences according to the presence of NGT/PEG (*p* > 0.05).

Regarding pharmacological treatment, patients medicated with dexamethasone showed a significantly more pronounced reduction in MNA values (M = −5.31) compared to those not medicated (M = −3.19), a statistically significant difference (t(119) = 3.714; *p* = 0.001). Patients medicated with mirtazapine, ondansetron, or metoclopramide showed greater mean reductions in MNA values, although these differences did not reach statistical significance (mirtazapine: *p* = 0.542; ondansetron: *p* = 0.226; metoclopramide: *p* > 0.05).

Patients who received psychological support presented higher MNA difference values, with the difference being statistically significant (t(119) = 3.541; *p* < 0.001).

The differences in weight, fatigue, and anorexia values according to rehabilitation were not statistically significant (*p* > 0.05).

A statistically significant association was observed between the presence of dysphagia and the absence of psychological support: 62.7% of patients without psychological support had dysphagia, compared to 28.8% among those who benefited from such support (Fisher’s test, *p* = 0.001). Despite this, the relationship between rehabilitation and dysphagia was not statistically significant (Fisher’s test, *p* = 0.154), and neither were the associations between psychological support and the presence of NGT/PEG (*p* = 0.091), between psychological support and xerostomia (*p* = 0.715), and between rehabilitation and xerostomia (*p* = 0.154).

By contrast, a statistically significant association was found between rehabilitation and the presence of a nasogastric tube or gastrostomy (NGT/PEG), with a significantly higher proportion of patients with rehabilitation among those with NGT/PEG (70.0%) compared to those without (19.7%) (Fisher’s test, *p* = 0.001). Additionally, the proportion of patients with taste alterations was significantly higher in the rehabilitation group (45.5%) when compared to the group without rehabilitation (22.5%) (Fisher’s test, *p* = 0.035). On the other hand, no significant association was found between psychological monitoring and the presence of taste alterations (Fisher’s test, *p* = 0.840).

## 4. Discussion

Oncological diseases remain the leading indication for referral to palliative care (*n* = 61; 47.7%), although there is a growing trend towards the inclusion of non-oncological pathologies, as evidenced by other studies [3,20,21]. Frailty syndrome was identified in 30 patients (23.5%), revealing a complex condition increasingly recognized as indicative of the need for palliative intervention, especially in the home context, which is in agreement with other international studies [16,22]. Dementia was the primary diagnosis of 19 patients (14.8%), constituting a progressively more prevalent cause of the need for palliative care. This data reflects, on the one hand, the aging of the population and, on the other, the more prolonged survival associated with neurodegenerative diseases, aspects widely described in current evidence [20,23,24]. Finally, heart failure was the main diagnosis of 18 patients (14.1%). This percentage aligns with the literature, which recognizes this pathology as one of the main non-oncological conditions that can benefit from a palliative approach [9,25,26].

The analysis reinforces the trend toward diversification of diagnoses covered by home palliative care, as observed by Ribeiro et al. (2024) and Magalhães et al. (2025) [16,21].

There was no significant difference between the number of days of follow-up and the associated pathology. On the one hand, the times of referral to home palliative care tend to be late, as reported in other international studies [20,27]. On the other hand, some patients had complex palliative needs for short periods, being discharged to the teams that had previously referred them, as described by Ribeiro et al. (2024) and Magalhães et al. (2025) [16,21].

In any case, it is important to emphasize that late referral to PC compromises the capacity for structured intervention, particularly in addressing ACS, optimizing quality of life, and supporting families in advanced care planning [28,29].

Our multivariable analysis revealed that a higher initial nutritional status, as measured by MNA, and use of enteral nutrition were paradoxically associated with an increased risk of significant weight loss, reflecting the severity of underlying conditions in these patient subgroups. In contrast to the weight loss model, the anorexia model indicated that a lower initial functional status and use of enteral nutrition were linked to a greater risk of subsequent anorexia, suggesting different underlaying models.

In the present study, no statistically significant correlation was identified between age at admission and initial body weight or BMI. This result is consistent with recent literature, which shows that, in the context of palliative care, especially in elderly populations with a high burden of morbidity, BMI and body weight tend to reflect not only physiological aging but above all the effects of advanced disease and the multifactorial mechanisms of Anorexia–Cachexia Syndrome [7,30,31]. Additionally, the lack of correlation reinforces the notion that BMI alone does not constitute a sensitive marker for assessing nutritional status or prognosis in patients undergoing specialized palliative care, and that a comprehensive and multifactorial nutritional assessment is necessary [1,10,32].

The data from this study demonstrated that there is no significant correlation between the age of patients (both at admission and at the time of death) and weight loss or decrease in BMI during follow-up. This result reinforces the need to consider the patient’s overall functionality and the moment in life in the expected disease trajectory when designing intervention strategies to mitigate cachexia and malnutrition in palliative care, as had already been emphasized by Cruz-Jentoft (2019) in the revised European Consensus on Sarcopenia, which highlights the importance of a comprehensive assessment of functionality and nutritional status in vulnerable geriatric populations [33].

Unlike what was observed in the initial fatigue, the differences in the final fatigue were statistically significant, with two groups standing out in particular. Patients with oncological pathologies presented higher values of final fatigue, which is consistent with the hypermetabolic and often aggressive nature of terminal oncological diseases. This accentuated fatigue may be a consequence of previous debilitating treatments, the presence of ACS, and a marked systemic inflammatory state [34,35]. On the other hand, patients with dementia showed lower levels of final fatigue. This phenomenon may be explained by slower and less abrupt trajectories of functional decline, reduced capacity for verbalization or subjective perception of the symptom due to cognitive deterioration, and by distinct pathophysiological mechanisms, in which cachexia may not assume such a marked expression as in neoplastic pathologies [23,24].

In our study, no statistically significant correlation was found between age (at admission and death) and initial and final anorexia levels. This finding is consistent with what is described by Argilés et al. (2014) and Fearon et al. (2012) on the pathophysiology of anorexia in the context of ACS, where the underlying mechanism is predominantly mediated by neuroendocrine and pro-inflammatory changes, such as increased pro-inflammatory cytokines (TNF-α, IL-6) and decreased ghrelin and other orexigenic stimuli—processes that are not dependent on chronological age [5,36].

Regarding the comparison of the initial PPS across different underlying pathologies, no statistically significant differences were observed. This result reinforces the notion that, regardless of the primary diagnosis (oncological disease, dementia, heart failure, or frailty syndrome), patients referred for home palliative care tend to have an advanced degree of functional limitation already, often being in low stages of PPS. These data are in agreement with Ribeiro et al. (2024), who assumed as a criterion for referral to the community palliative care support team a PPS below 60%, and with Bischoff et al. (2024), who demonstrated that patients with different diagnoses often have a low PPS at the time of referral for palliative care [21,37].

No significant correlation was found between weight, BMI, and fatigue. Although weight loss and a decreased BMI indicate nutritional deterioration, they are not directly related to fatigue, which is caused by a complex set of factors. There was, however, a negative correlation between initial anorexia and final BMI, indicating that greater anorexia at the beginning leads to a lower final BMI. In addition, there was a positive relationship between initial anorexia and initial fatigue and between final anorexia and final fatigue, the latter being stronger. These results suggest that greater anorexia, both at the beginning and at the end, is associated with higher levels of fatigue, especially in the terminal phase, aggravated by physiological deterioration as described by Stewart et al. (2008), Baracos et al. (2018), and Solheim et al. (2018) [2,31,35].

Our study revealed that patients without dysphagia had a lower initial weight (mean of 59.62 kg) compared to those with dysphagia (65.31 kg). Despite this, patients with dysphagia showed a greater percentage of weight loss throughout treatment (9.95% versus 4.43%). Dysphagia is an important risk factor for malnutrition, leading to rapid weight loss and worsening of global functionality. Furthermore, patients with dysphagia had higher levels of final fatigue and anorexia, indicating greater physical deterioration and impact on quality of life. Dysphagia, as a sign of organ failure, limits food and water intake, aggravating malnutrition, fatigue, and dehydration [10,38].

In our study, patients with NGT/PEG experienced greater weight loss (11.28%) compared to those without enteral nutrition (5.62%), contrary to the expectation that nutritional support would preserve weight. This divergence is explained by the fact that, in advanced stages, enteral nutrition does not prevent weight loss as described by several authors [16,24,39,40]. In patients with advanced cachexia, particularly those with dementia, enteral nutrition may not improve survival or halt nutritional decline, especially if initiated at a refractory stage [41]. This could be due to selection bias, where artificial nutrition is initiated when weight loss is already evident, or reverse causality. As such, it is paramount to balance, individualize, and proportion support modalities with each patient’s trajectory, a sentiment reinforced by ASCO (2020) in their guidelines [42,43].

In addition, patients with NGT/PEG experienced less final fatigue, possibly due to reduced effort in feeding and improved symptomatic control. On the other hand, patients with taste alterations lost more weight (10.27%), which indicates that dysgeusia and ageusia negatively affect food intake, aggravating nutritional deterioration in terminally ill patients. Holm et al. (2007) [40] highlight that while PEG placement aims to improve nutritional status, its effectiveness in advanced palliative care settings is often limited by factors beyond caloric intake. These include the underlying catabolic state driven by inflammatory cytokines, reduced anabolism due to tumor burden or organ failure, and the patient’s declining functional capacity to utilize nutrients effectively. Therefore, while enteral nutrition may alleviate some symptoms and provide a sense of comfort, it may not fully counteract the complex metabolic derangements contributing to weight loss in advanced Anorexia–Cachexia Syndrome [40]. Furthermore, the decision to initiate enteral nutrition often reflects a recognition of advanced disease, potentially skewing the results towards a population with a poorer overall prognosis.

Our study highlights the central role of often underestimated symptoms like xerostomia, dysgeusia, and dysphagia. These oral manifestations have a substantial impact on reduced food intake, nutritional compromise, and quality of life, as documented in studies particularly in head and neck oncology [44,45]. We advocate for a multimodal therapeutic approach that combines pharmacological interventions (e.g., artificial saliva), nutritional support (e.g., texture-modified diets), and speech therapy to address these symptoms proactively.

The present study demonstrated that weight loss was significantly higher in patients with xerostomia, with an average of 9.66%, compared to 5.02% in patients without this complaint (MU = 229,000, *p* = 0.012). This population also presented significantly higher values of final fatigue and final anorexia (4.52 vs. 3.06; MU = 1,395,000; *p* = 0.034). These findings reinforce the understanding of xerostomia as a precipitating and aggravating factor of the terminal cachectic cascade, promoting a negative synergy between interdependent symptoms such as anorexia and fatigue [34,35,46,47].

A significantly higher prevalence of dysphagia was observed in patients with xerostomia, corroborating the existing evidence on the interrelationship between alterations in salivation and swallowing function, as reported by Baijens et al. (2016) and Davies et al. (2000) [38,48]. This functional interdependence highlights the need for early assessment and coordinated intervention to maintain food safety, preventing aspiration and promoting overall comfort [38,48,49].

Our analysis of functional outcomes measured through the Palliative Performance Scale (PPS) is further strengthened by citing studies that confirm its prognostic validity. Prompantakorn et al. (2021) demonstrated that the PPS is a reliable indicator of survival in both cancer and non-cancer patients, providing essential support for advance care planning and resource allocation in home settings [50]. In our study, both the initial PPS and the one-week post-intervention PPS showed significant negative correlations with final weight, fatigue, and anorexia, confirming their value as prognostic markers of functional reserve and global clinical status. Higher PPS values were associated with less functional impairment and less symptomatic expression, while their progressive reduction reflected the progression of organ failure. Patients with taste alterations and xerostomia had higher PPS values at admission. Similarly, PPS at the first week was higher among patients with xerostomia. These data indicate that such symptoms may emerge in functional stages that are still relatively preserved, preceding more severe manifestations such as severe dysphagia and refusal to eat [38,48,49].

Similarly, the Mini Nutritional Assessment (MNA) emerges as a crucial prognostic tool. A systematic review has shown that low MNA scores correlate with mortality, disease progression, and worsening quality of life [51]. The integration of PPS and MNA emerges as a valid strategy for identifying frail patients who could benefit from personalized home support pathways.

The MNA did not correlate with age or sex but varied according to diagnosis: cancer and dementia patients had higher scores than frailty patients, reflecting different clinical trajectories. There was a positive and moderate correlation between the initial MNA and that after one week of follow-up by the team, indicating some short-term stability, although cancer patients showed greater decline over time, which is in line with what was described by Poisson et al. (2021) and by Solheim et al. (2018) in the context of nutritional assessment in advanced disease [30,31].

Patients with dysphagia had a higher initial MNA, possibly due to early intervention, while those with taste alterations and xerostomia had a lower final MNA, impairing intake.

Dexamethasone was associated with a deterioration in MNA, indicating more severe cases, as observed by Del Fabbro et al. (2006) [46]. In contrast, psychological support was associated with worsening nutritional evolution, possibly due to a greater clinical burden, in line with the conclusions of Hudson et al. (2015) [3].

Dysphagia was more common among patients without psychological support, and those undergoing rehabilitation showed greater use of NGT/PEG and taste alterations, highlighting the need for greater integration of symptomatic dimensions in multidisciplinary monitoring [3,40,46].

The concept of a multimodal approach to cachexia, including nutrition, physical activity, and inflammation management, is now well-established [42]. Recent clinical trials, such as MENAC (Multimodal Exercise, Nutrition, and Anti-inflammatory medication for Cachexia), demonstrate a slower rate of weight loss compared to standard care. Our study implicitly supports this direction, and we propose that a synergistic care plan, targeting multiple aspects of ACS, is essential for optimal outcomes.

It should be noted that the exclusion of patients with follow-up periods shorter than 30 days, while methodologically justified, may have resulted in the omission of cases with rapid clinical deterioration. This could lead to an underestimation of the acute impact and severity of ACS in the most fragile or terminal patients. The late recruitment into the palliative care network observed in our study is consistent with evidence showing that earlier referrals are associated with less aggressive care, greater alignment with patient wishes, and improved quality of life [52,53]. This underscores the need for increased awareness and proactive referral pathways to palliative care to maximize the benefits of early intervention in ACS.

## 5. Limitations of This Study

The retrospective nature of this study inherently limits the ability to make causal inferences. Data were extracted from existing clinical records, which may lack uniformity in symptom documentation, intervention timing, and follow-up assessments. This design restricts the ability to establish temporal relationships and introduces the potential for selection bias and information bias. This study is based on a single community palliative care team, which may limit the external validity and generalizability of the findings. Sociodemographic, institutional, and cultural factors specific to the studied region may not reflect broader national or international contexts in palliative care. Additionally, the sample is relatively small, and the population exhibits heterogeneity. Although the cohort comprises 128 patients, subgroup analyses (e.g., those with PEG/NGT, xerostomia, or psychological support) are based on small numbers, which compromises their statistical power and increases the risk of type II errors. The clinical heterogeneity of the sample (including cancer, frailty, dementia, and heart failure) also complicates the interpretation of aggregated outcomes.

The association between dexamethasone use and the worsening of nutritional status should be interpreted with caution due to potential confounding by indication. Patients prescribed dexamethasone may have been in a more advanced stage of disease or experiencing more severe symptoms, such as uncontrolled nausea or pain, which could independently contribute to nutritional decline. While we adjusted for some factors in our multivariable analysis, residual confounding cannot be entirely ruled out. Moreover, our study relied on the recording of subjective symptoms such as fatigue and anorexia, which are inherently susceptible to reporting bias. These symptoms were assessed by medical doctors and nurses based on patient self-reports, which may be influenced by factors such as recall bias, cultural background, or individual pain thresholds. While we attempted to mitigate this bias by using a standardized assessment scale, the potential for subjective bias remains a limitation.

## 6. Conclusions

This study provides a comprehensive view of the complex interplay of factors affecting nutritional and functional status in home-based palliative care.

Our findings underscore the importance of early identification and proactive management of often-underestimated symptoms like dysphagia, xerostomia, and taste alterations.

Multimodal interventions, tailored to the individual patient’s trajectory rather than solely dictated by the underlying diagnosis, are crucial.

While enteral nutrition may play a role, its impact should be carefully considered in light of the advanced disease stage.

Moving forward, these findings suggest the need for clinical protocols in home palliative care that prioritize early assessment of oral symptoms and regular monitoring of nutritional and functional status using tools like the MNA and PPS, enabling timely intervention and personalized care plans to optimize quality of life and autonomy for patients with advanced illness.

## Figures and Tables

**Figure 1 jcm-14-06167-f001:**
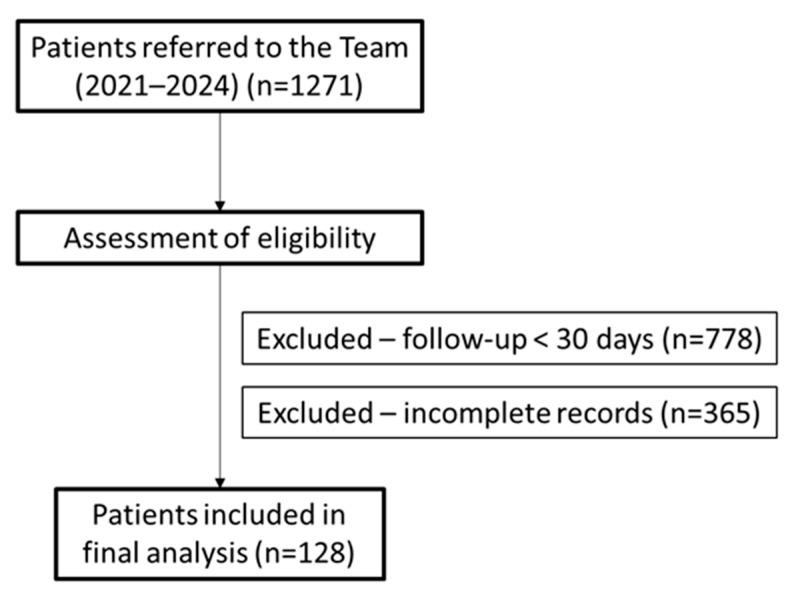
Flow diagram of patient inclusion and exclusion in the retrospective cohort, based on STROBE guidelines. *n*: number.

**Table 1 jcm-14-06167-t001:** Sociodemographic assessment of the patients that constituted the study sample.

**Sex**	** *n* **	**%**
Female	67	52.3
Male	61	47.7
**Main pathology**	** *n* **	**%**
Dementia	19	14.8
Heart failure	18	14.1
Oncologic disease	61	47.7
Frailty	30	23.5
	**Average ± SD**	
Age at admission	82.7 ± 14.4	
Age of death	82.5 ± 14.4	

*n* = number; % = percentage; SD = standard deviation.

**Table 2 jcm-14-06167-t002:** Multivariable proportional hazard regression analysis of factors associated with risk of significant weight loss (>5%).

Weight Loss > 5%	Adjusted Odd Ratios	95% Confidence Interval	*p* Value
Gender—Male	0.061	0.004–1.031	0.53
MNA first	3.797	1.473–9.788	0.006
MNA first week plus medicine	0.282	0.120–0.660	0.004
NGT or PEG	7.418	1.506–36.525	0.014

% = percentage; *p* = significance; MNA = Mini Nutritional Assessment; NGT: nasogastric tube; PEG = percutaneous endoscopic gastrostomy.

**Table 3 jcm-14-06167-t003:** Multivariable proportional hazard regression analysis of factors associated with subsequent anorexia.

Anorexia After	Adjusted Odd Ratios	95% Confidence Interval	*p* Value
PPS first/initial	0.069	0.031–0.107	<0.001
MNA first week plus medicine	−0.404	−0.517–−0.290	<0.001
Psychology	−1.578	−2.640–−0.517	0.04
NGT or PEG	−0.890	−1.555–−0.224	0.009

% = percentage; *p* = significance; MNA = Mini Nutritional Assessment; NGT: nasogastric tube; PEG = percutaneous endoscopic gastrostomy.

**Table 4 jcm-14-06167-t004:** Comparison of nutritional and symptomatic parameters according to the presence of dysphagia, use of NGT/PEG, taste alterations, and xerostomia.

	Initial Weight (A ± SD)	Weight Loss (A ± SD)	Initial Fatigue (A ± SD)	Final Fatigue (A ± SD)	Initial Anorexia (A ± SD)	Final Anorexia (A ± SD)
**Patients without dysphagia**	**59.62 ± 10.45**	**4.43% ± 4.61%**	6.39 ± 1.56	**5.31 ± 1.62**	5.31 ± 2.14	**3.08 ± 2.07**
**Patients with dysphagia**	**65.31 ± 12.24**	**9.95% ± 6.8%**	5.94 ± 1.55	**5.90 ± 2.42**	5.52 ± 1.82	**4.54 ± 3.23**
** *p* **	**0.027**	**0.002**	0.085	**0.042**	0.136	**0.047**
**Patients without NGT/PEG**	62.61 ± 12.88	**5.62% ± 6.37%**	5.87 ± 1.35	**5.95 ± 1.8**	4.88 ± 2.38	3.94 ± 2.83
**Patients with NGT/PEG**	61.13 ± 10.4	**11.28% ± 5.94%**	5.7 ± 1.75	**4.90 ± 2.36**	5.84 ± 1.17	3.16 ± 2.73
** *p* **	0.678	**0.005**	0.791	**0.038**	0.092	0.291
**Patients without change in taste**	62.28 ± 11.84	**4.98% ± 5.7%**	6.06 ± 1.66	**5.31 ± 2.06**	5.22 ± 2.11	3.45 ± 2.6
**Patients with change in taste**	62.35 ± 11.2	**10.27%± 5.71%**	6.58 ± 1.25	**6.24 ± 1.71**	5.88 ± 1.63	4.3 ± 2.9
** *p* **	0.496	**0.002**	0.251	**0.030**	0.229	0.183
**Patients without xerostomia**	62.64 ± 11.41	**5.02% ± 5.23%**	6.16 ± 1.61	**5.17 ± 1.83**	5.43 ± 2.05	**3.06 ± 2.08**
**Patients with xerostomia**	61.74 ± 12.1	**9.66% ± 6.77%**	6.25 ± 1.53	**6.08 ± 2.13**	5.35 ± 1.97	**4.52 ± 3.18**
** *p* **	0.739	**0.012**	0.755	**0.009**	0.593	**0.034**

*n* = number; % = percentage; A = average; SD = standard deviation; *p* = significance; Bold = statistically significant.

**Table 5 jcm-14-06167-t005:** Comparison of initial PPS values and those at the 1st week according with anthropometric parameters and symptoms of anorexia, fatigue, taste alterations, and xerostomia.

	Initial PPS (A ± SD)	PPS After 1 Week (A ± SD)
**Initial weight**	−0.035	**−0.269**
**Final weight**	**−0.281**	−0.262
**BMI**	0.093	−0.160

**Initial fatigue**	**−0.201**	**−0.312**
**Final fatigue**	0.132	**−0.252**
**Initial anorexia**	**−0.285**	−0.161
**Final anorexia**	0.155	−0.050
**Without change in taste**	**25.16 ± 12.5**	12.33 ± 6.19
**With change in taste**	**29.39 ± 12.98**	12.73 ± 5.74
** *p* **	**0.023**	0.493
**Without xerostomia**	**23 ± 9.53**	**11.3 ± 4.17**
**With xerostomia**	**30.56 ± 14.97**	**13.89 ± 7.63**
** *p* **	**0.001**	**0.020**

A = average; SD = standard deviation; *p* = significance; Bold = statistically significant; PPS = Palliative Performance Scale; BMI = body mass index.

**Table 6 jcm-14-06167-t006:** Correlations between initial and final MNA (Mini Nutritional Assessment) values and nutritional and symptomatic parameters.

	Initial MNA	Final MNA
**Initial weight**	0.271	0.181
**Final weight**	**0.261**	**0.436**
**Weight loss**	−0.001	**−0.415**
**Initial BMI**	**0.474**	**0.261**
**Final BMI**	**0.534**	**0.636**
**Initial fatigue**	**−0.211**	**−0.340**
**Final fatigue**	−0.101	**−0.337**
**Initial anorexia**	**−0.423**	**−0.365**
**Final anorexia**	**−0.320**	**−0.467**

MNA = Mini Nutritional Assessment; BMI = body mass index; Bold = statistically significant.

**Table 7 jcm-14-06167-t007:** Initial and 1st week MNA (Mini Nutritional Assessment) values according to diagnosis.

	Initial MNA (A ± SD)	MNA After 1 Week (A ± SD)
**Dementia**	22.11 ± 4.28	19.42 ± 5.71
**Heart failure**	21.88 ± 3.33	19.35 ± 4.66
**Oncological disease**	22.66 ± 3.34	17.10 ± 3.99
**Frailty**	20.04 ± 4.02	16.30 ± 3.96

A = average; SD = standard deviation; MNA = Mini Nutritional Assessment.

## Data Availability

Upon reasonable request.

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
