# Peer review of "Management of Anorexia–Cachexia Syndrome in a Community Palliative Care Support Team"

_jcm, 2025, doi:10.3390/jcm14176167_

Round 1

Reviewer 1 Report

Comments and Suggestions for Authors

The objective of the manuscript, "Management of Anorexia-Cachexia Syndrome in a Community Palliative Care Support Team," is to "evaluate the prevalence, management strategies, and outcomes of patients in a home-based palliative care team."
The article therefore addresses a topic that is still underexplored and very interesting: patients at home, rather than hospitalized patients, who have been more widely studied to date. Importantly, the sample also includes non-cancer patients.
However, I suggest some improvements of varying degrees of importance.
Lines 66-68: Is ACS really relevant for the survival prognosis of patients undergoing palliative care and therefore at the end of life?
Figure 1: I suggest improving the flow chart by adding the numbers of patients excluded for various reasons and structuring the figure differently (the arrow that starts from "patients referred..." and passes through excluded to arrive at included).
Regarding the methodology, it is necessary to explain how fatigue was assessed (tools, cutoffs, etc.). The PPS should also be explained in a few sentences how it works.
In the results, I believe it would be useful to indicate the ages of the cancer patients and those with, for example, dementia, because I believe they are different and, above all, I wonder if it was correlated with the outcomes.
The weak point concerns the management, which is indicated as one of the objectives of the study: How was it decided who would start artificial nutrition and, moreover, what did those who were not receiving enteral nutrition do (free diet, diet for dysphagic patients?).
The results contain too many comparisons, making it difficult to follow the text. I suggest:
a) respecting the objectives in the presentation of the results (prevalence, management, and outcomes), giving each one equal importance.
b) The comparisons made are too "unpackaged." It makes no sense to isolate symptoms and study their correlation with a given outcome separately (for example, those with a low BMI may also have dysphagia and anorexia, etc.). For this reason, multivariate analysis absolutely must be introduced!
I believe that after appropriate revisions and modifications, the manuscript will be very interesting!

Author Response

Replies to the reviewers

We thank all reviewers for their comments and suggestions. Their suggestions improved the quality of the manuscript.

Please find our replies below.

Reviewer 1:

1- Lines 66-68: Is ACS really relevant for the survival prognosis of patients undergoing palliative care and therefore at the end of life?

Answer: We thank the reviewer for raising this important point about the relevance of ACS in palliative care. We acknowledge that, in the advanced stages of palliative care, extending survival is often not the primary objective. However, we believe that managing ACS remains critically important for improving patients' quality of life. ACS significantly impacts physical function, symptom burden (such as fatigue and weakness), psychological well-being, and the ability to enjoy social interactions and food-related activities. [http://dx.doi.org/10.1016/j.jpainsymman.2016.12.325; https://doi.org/10.1016/j.mcna.2016.04.020; doi: 10.1007/s13539-014-0142-1; DOI: 10.1097/00152192-200507000-00012]. Even modest improvements in these areas can substantially enhance the patient's experience in their remaining time. While the impact on survival may be less direct, improved nutritional status and symptom control may, in some cases, contribute to a slightly more comfortable or prolonged survival. We made some changes in those lines. Now you could read: “In palliative care, Anorexia-Cachexia Syndrome (ACS) is a particularly relevant concern, affecting a significant proportion of patients with advanced chronic illnesses (1,2). While the impact on overall survival in this population may be complex, ACS profoundly affects quality of life. The syndrome's characteristics – involuntary weight loss, anorexia, and metabolic changes – lead to significant functional decline, increased symptom burden (e.g., fatigue, weakness), and a diminished ability to participate in daily activities and social interactions. These factors contribute to a substantial worsening of the psychological well-being of both patients and their caregivers (2-4).”

2- Figure 1: I suggest improving the flow chart by adding the numbers of patients excluded for various reasons and structuring the figure differently (the arrow that starts from "patients referred..." and passes through excluded to arrive at included).

Answer: We thank the reviewer for the suggestion to improve Figure 1. We have revised the flow chart to provide greater clarity regarding the patient selection process. Specifically, we have:

  • Restructured the flow to clearly show the branching of patients into exclusion categories.
  • Included the number of total patients (n=1271) and number of patients excluded for each reason (follow-up less than 30 days: n=778; incomplete records: n=365).

The revised figure provides a more transparent representation of the study population.

3- Regarding the methodology, it is necessary to explain how fatigue was assessed (tools, cutoffs, etc.). The PPS should also be explained in a few sentences how it works.

Answer: "We thank the reviewer for pointing out the need for more detail on our assessment methods. We have added the following information to the methodology section:

  • Anorexia and fatigue Assessment:

Anorexia and fatigue were assessed using a numerical rating scale ranging from 0 to 10, where 0 represents "no fatigue" and 10 represents "worst possible fatigue". Patients were asked to rate their current level of fatigue at the time of each assessment. A score of 4 or higher was considered to indicate clinically significant fatigue. Anorexia and fatigue were assessed by nurses at the time of the initial assessment and at the end of the first week of follow-up. Whenever we could not rely on self-reporting, we apply a peer-review scale and consolidate results into three groups (mild, moderate, or severe), corresponding to values 1-3, 4-7, and 8-10. Using a weighted average, to include all data, we considered 2, 5.5 and 9 as numerical reference values. Statistical analysis did not reveal significant confounding in converting descriptive categories to numerical ones.”

  • Palliative Performance Scale (PPS):

“Functional status was assessed using the Palliative Performance Scale (PPS) (11), a widely used tool in palliative care that assesses a patient's ability to ambulate, perform activities of daily living, level of consciousness, and nutritional intake. The PPS ranges from 0% (bedridden, total care required) to 100% (fully functional and independent). PPS scores were determined by nurses and medical doctors based on their clinical assessment of the patient.”

4- In the results, I believe it would be useful to indicate the ages of the cancer patients and those with, for example, dementia, because I believe they are different and, above all, I wonder if it was correlated with the outcomes.

Answer: We thank the reviewer for suggesting that we examine the role of age in our study. We agree that age is a potentially important confounding factor and have now analyzed the age distribution across the different diagnostic groups.

We added and highlighted this sentence to the manuscript: “The mean age at admission varied across diagnostic groups, but without statistically significance (p > 0.05). Patients with Dementia were, on average, slightly older (85.47±11.51 years) compared to patients with Cancer (83.38±10.74 years), Frailty Syndrome (83.00±9.48 years) and Heart Failure (84.67±8.76 years).” (pages 6-7)

5- The weak point concerns the management, which is indicated as one of the objectives of the study: How was it decided who would start artificial nutrition and, moreover, what did those who were not receiving enteral nutrition do (free diet, diet for dysphagic patients?).

Answer: We thank the reviewer for raising this important question about our approach to nutritional management. We have added and highlighted detail to the methodology section (page 5) to clarify our decision-making process.

  • Protocol for Nutrition Decisions:

“Decisions regarding artificial nutrition were made on a case-by-case basis through a multidisciplinary team assessment that considered the patient's overall disease trajectory, global functionality (as measured by PPS), estimated survival expectancy, and the degree and quickness of decline in oral intake. In accordance with Magalhães et al (2025) (19), percutaneous endoscopic gastrostomy (PEG) was considered for patients with a survival expectancy of more than 4 weeks and an anticipated prolonged inability to maintain adequate oral intake. For patients experiencing a sudden loss of oral intake requiring immediate nutritional support, a nasogastric tube (NGT) was placed as a temporary measure while awaiting PEG placement. PEG placement typically occurred within 1-2 weeks, taking into consideration the patient's expected survival benefit and the balance of risks and benefits. All decisions were made in consultation with the patient (if able) and their family, respecting their values and preferences.”

  • Management of Patients Not Receiving Enteral Nutrition:

“For patients who did not receive enteral nutrition, management focused on optimizing oral intake through a combination of strategies. Patients were offered a "free diet" of preferred foods, as tolerated. Dietary modifications, such as pureed foods or thickened liquids, were used to address dysphagia. Oral nutritional supplements were provided to augment caloric intake. In some cases, the focus shifted to "comfort feeding," prioritizing the patient's enjoyment of food, even if nutritional goals were not fully achieved. Symptom management, including interventions for nausea, pain, and xerostomia, was a key component of care.”

6- The results contain too many comparisons, making it difficult to follow the text. I suggest:
a) respecting the objectives in the presentation of the results (prevalence, management, and outcomes), giving each one equal importance.
b) The comparisons made are too "unpackaged." It makes no sense to isolate symptoms and study their correlation with a given outcome separately (for example, those with a low BMI may also have dysphagia and anorexia, etc.). For this reason, multivariate analysis absolutely must be introduced!

Answer: We thank the reviewer for identifying the need for a more structured presentation of our results and for highlighting the importance of accounting for the interrelationships among variables. We have now:

  • Restructured the Results section to align with our study objectives (prevalence, management, outcomes).
  • Performed multivariate analyses to examine the independent predictors of our key outcomes. We added and highlighted in the methodology “Multivariable logistic regression analysis was performed to identify independent clinical and demographic factors associated with weight loss and anorexia, considering the other variables. The predictive model was created using logistic and linear regression analysis and dimension reduction through the backward feature elimination method. Variables with clinical relevance included in the multivariable analysis were all found to be associated with the outcomes in the univariable analysis at a significance level of P < 0.25. To ensure that potentially relevant clinical factors were not excluded from the multivariable analysis, a less stringent significance level of p < 0.25 was used as a threshold for variable inclusion in the initial model. This approach is recommended in situations where there is a desire to explore a wider range of potential predictors.”
  • Revised the Discussion section to emphasize the importance of considering multiple factors simultaneously and to acknowledge the limitations of our study.

We believe that these changes have significantly strengthened the manuscript and provide a more nuanced understanding of the factors influencing outcomes in patients with ACS.

Reviewer 2 Report

Comments and Suggestions for Authors

Peer Review Report for Manuscript “Management of Anorexia-Cachexia Syndrome in a Community Palliative Care Support Team” (jcm-3794310)

Dear Authors,

Thank you for the opportunity to review your manuscript entitled Management of Anorexia-Cachexia Syndrome in a Community Palliative Care Support Team. This retrospective cohort study addresses a clinically relevant and often underexplored topic in palliative medicine: the prevalence, management strategies, and clinical and patient-centred outcomes of Anorexia-Cachexia Syndrome (ACS) within home-based palliative care. Drawing on the records of 128 patients followed between 2021 and 2024, the study investigates the interrelationships among nutritional decline, fatigue, anorexia, oral symptoms, and functional deterioration, compares oncological and non-oncological diagnoses, and explores the impact of interventions such as enteral nutrition, corticosteroids, and psychological support. The work aligns well with the scope of the Journal of Clinical Medicine and offers potentially valuable insights for community palliative care teams.

The manuscript’s strengths include its integration of oncological and non-oncological conditions, coverage of multiple symptom domains, and comprehensive ethical compliance. The dataset is rich, and the authors present an array of statistical analyses spanning correlations, group comparisons, and symptom associations. The emphasis on symptoms often underestimated in clinical practice—xerostomia, dysgeusia, and dysphagia—is commendable and may encourage more holistic care approaches.

Nonetheless, several issues currently limit the clarity, methodological transparency, and interpretative depth. While the descriptive reporting is thorough, some sections contain unnecessary repetition of identical findings, which detracts from concision and readability. Moreover, certain methodological choices—such as the selection of statistical tests for specific comparisons—are not sufficiently justified, and the rationale underpinning the study objectives is occasionally vague. The Discussion, while clinically rich, would benefit from less reiteration of results already presented in the Results and from a stronger focus on interpreting unexpected findings within the broader evidence base.

Title and Abstract

The title is accurate and clearly reflects the central objective of the study. However, although the abstract outlines the main objectives, I suggest restructuring it according to the following criteria (https://www.mdpi.com/journal/jcm/instructions):

- Background/Objectives (state the purpose, research questions, and context),

- Methods (describe the study design, main variables, participant characteristics, and whether the study was registered, keeping in mind that clinical trials must include the elements required by CONSORT),

- Results (report the main results with key statistics),

- Conclusions (summarize the main interpretations and implications for clinical practice). I suggest this because, given the empirical nature of this study, I believe the structured format facilitates the reader's understanding.

Introduction

The introduction correctly defines Anorexia-Cachexia Syndrome as a complex, multifactorial, and highly prevalent condition in patients with advanced chronic diseases. The authors provide updated definitions and emphasize their prognostic relevance, citing landmark studies. This approach is appropriate and demonstrates familiarity with the international scientific debate.

However, the text offers room for improvement in terms of clarity of objectives and the transition from a general framework to a specific focus. The stated objective of "identifying patterns" appears overly general: it is unclear whether it refers to clinical, nutritional, biochemical, therapeutic patterns, or a combination thereof. It would be helpful to clarify this definition from the introduction, so as to allow the reader to fully understand the scope and direction of the analysis.

The cited literature adequately covers the pathophysiological aspects of ACS, but the rationale for focusing on the home setting is not fully developed. In this area, the available literature is fragmented and often limited to hospital-based oncology settings. Further clarifying the paucity of studies in the community-based setting and briefly comparing the operational differences compared to inpatient care would strengthen the study's justification.

Furthermore, the introduction of this approach could benefit from greater attention to the organizational and sociocultural aspects that influence home management of ACS: caregiver training, available resources, multidisciplinary coordination, and logistical barriers. These factors are crucial to the effectiveness of interventions and could also be discussed in light of similar experiences in other clinical areas. For example, the literature on pain management in palliative care has shown that the use of integrated scales (Mercadante, S., Lo Cascio, A., Adile, C., Ferrera, P., & Casuccio, A. (2023). Maddalena Opioid Switching Score in patients with cancer pain. Pain, 164(1), 91–97. https://doi.org/10.1097/j.pain.0000000000002669) improves the appropriateness and timeliness of therapeutic decisions. The parallel is pertinent: also for ACS, a structured and standardized approach could optimize management, especially when dealing with multiple and interconnected symptoms. Another aspect that could be developed concerns “minor” or underestimated symptoms, such as xerostomia (Murphy Dourieu, E., Lisiecka, D., Evans, W., & Sheahan, P. (2025). Xerostomia: a silent burden for people receiving palliative care - a qualitative descriptive study. BMC palliative care, 24(1), 1. https://doi.org/10.1186/s12904-024-01617-x) and dysgeusia, which the study treats in an innovative way (Ferrara, G., Palomares, S. M., Anastasi, G., Parozzi, M., Godino, L., Gazineo, D., Sguanci, M., & Mancin, S. (2025). Anosmia, dysgeusia and malnutrition in chronic kidney disease: A systematic review. Nefrologia, 45(2), 106–115. https://doi.org/10.1016/j.nefroe.2025.02.002). Including the hypothesis that proactive management of these symptoms can influence the clinical trajectory right from the introduction would provide an interpretative anchor to the results and strengthen the coherence of the argument.

Materials and Methods

The description of the study design is clear, but the specification "without intervention" is superfluous, as the retrospective observational nature of the study already implies the absence of experimental interventions. Eliminating this term would make the text more fluent and direct.

The section on inclusion and exclusion criteria is well-structured, but raises a relevant question: among the diagnoses considered, chronic kidney disease (CKD), a condition often associated with cachexia in palliative care, is not included. It would be appropriate to specify whether CKD was absent from the cohort, undetected, or deliberately excluded, and to justify the choice, as its inclusion could have enriched the comparative analysis.

The description of the collected variables (lines 104–113) is exhaustive, but the presentation in a single, very long sentence compromises readability. Grouping the data into thematic subcategories—sociodemographic, clinical, functional, symptomatic, and therapeutic—would enhance clarity and allow the reader to better navigate the flow of information.

The list of statistical tests used (lines 123–134) is also comprehensive, but there is a lack of correspondence between each test and the specific hypothesis or variable analyzed. Explicitly stating, for example, that the Student t-test was used to compare means of two groups in continuous variables, while the chi-square test was used to analyze categorical distributions, would improve methodological transparency and replicability.

Looking ahead, the manuscript could benefit from a flowchart illustrating not only recruitment but also the distribution of the main variables, thus providing a quick overview of the dataset.

Results

The results are rich and detailed, but contain redundancies that make them difficult to read. The correlation between baseline anorexia and baseline fatigue (r = 0.459; p = 0.001) is reported twice (lines 155 and 166), as is the association between dysphagia and xerostomia (lines 191 and 202–203), and the correlation between initial MNA and MNA at one week (lines 236–237 and 268–269). Eliminating repetition would make the section more streamlined.

The tables are generally clear, but some include overlapping parameters that could be merged. The text is predominantly descriptive; adding brief interpretative comments immediately following key associations would help maintain the reader's attention and connect the data to clinical practice. Of particular note is the counterintuitive finding that patients with NGT/PEG exhibit greater weight loss than those without enteral nutrition. This aspect, also central to the literature (Holm et al., 2007), should be highlighted more prominently in the results, anticipating the discussion of possible pathophysiological and clinical explanations and better aligned with recent international literature.

 Discussion

The manuscript's discussion is undoubtedly clinically relevant, as it addresses Anorexia-Cachexia Syndrome (ACS) in a little-explored home setting. The authors highlight the syndrome's transversal nature with respect to diagnoses, the central role of oral symptoms, and treatment pathways that often begin too late. However, to enhance the text's scientific validity, I suggest enriching it with references to recent literature.

The finding that patients receiving enteral nutrition (NGT/PEG) unexpectedly experience greater weight loss deserves in-depth analysis. The literature argues that in patients with advanced cachexia—particularly those with dementia—enteral nutrition neither improves survival nor halts nutritional decline, especially if initiated at an already refractory stage. This may be due to the selection of the most compromised patients or to reverse causality: the decision to administer artificial nutrition is often made when weight loss is already evident. Guidelines and prospective studies indicate that support modalities must be balanced, individualized, and proportionate to the patient's trajectory (Amano et al., 2021; ASCO, 2020).

Another distinguishing aspect of the study is its focus on symptoms that tend to be underestimated in clinical practice, such as dry mouth, dysgeusia, and dysphagia. Several studies, particularly in head and neck oncology, have documented their substantial impact on reduced food intake, nutritional compromise, and deterioration in quality of life (Bressan et al., 2016; Ghias, 2023). Including the importance of proactive management of these symptoms in the discussion and suggesting multimodal therapeutic approaches—combining pharmacological, nutritional, and speech therapy support—would enhance the originality of the research.

The concept of a multimodal approach to cachexia, including nutrition, physical activity, and inflammation management, is now well-established (Arends et al., 2021), and is the basis of recent clinical trials such as MENAC (Multimodal Exercise, Nutrition, and Anti-inflammatory medication for Cachexia), which show a slower rate of weight loss compared to standard care. Your study implicitly suggests this direction: a direct reference to this evidence would strengthen the implicit proposal for a synergistic care plan.

The analysis of results related to functional outcomes, measured through the Palliative Performance Scale (PPS), can be further strengthened by citing studies that confirm its prognostic validity. For example, Prompantakorn et al. (2021) demonstrated that the PPS is a reliable indicator of survival in both cancer and non-cancer patients, providing essential support for advance care planning and resource allocation in home settings.

Similarly, the importance of MNA as a prognostic tool should be highlighted. A systematic review has shown that low MNA correlates with mortality, disease progression, and worsening quality of life (Torbahn et al., 2020). The integration of PPS and MNA emerges as a valid strategy for identifying frail patients who could benefit from personalized home support pathways.

Finally, it should be noted that late recruitment into the palliative care network—also observed in the present study—is consistent with evidence showing that earlier referrals are associated with less aggressive care, greater alignment with patient wishes, and in many cases improved quality of life (Temel et al., 2014; Mukamel et al., 2023). Including this link would contribute to a more robust organizational framework.

  • Amano, K., et al. (2021). Effects of enteral nutrition and parenteral nutrition on survival in patients with advanced cancer cachexia: Analysis of a multicenter prospective cohort study. Clinical Nutrition, 40(3), 1168–1175. https://doi.org/10.1016/j.clnu.2020.07.027
  • Arends, J., et al. (2021). Cancer cachexia in adult patients: ESMO Clinical Practice Guidelines. Annals of Oncology, 32(12), 1231–1245. https://doi.org/10.1016/j.annonc.2021.12.004
  • ASCO (2020). Roeland, E. J., et al. Management of cancer cachexia: ASCO Guideline. Journal of Clinical Oncology, 38(20), 2432–2452. https://doi.org/10.1200/JCO.20.00611
  • Bressan, V., et al. (2016). The effects of swallowing disorders, dysgeusia, oral mucositis and xerostomia on nutritional status, oral intake and weight loss in head and neck cancer patients: A systematic review. Cancer Treatment Reviews, 45, 105–119. https://doi.org/10.1016/j.ctrv.2016.03.009
  • Ghias, K. (2023). The impact of treatment-induced dysgeusia on oral intake and nutritional status. Cancer Reports, e1792. https://doi.org/10.1002/cnr2.1792
  • Prompantakorn, P., Angkurawaranon, C., Pinyopornpanish, K., Chutarattanakul, L., Aramrat, C., Pateekhum, C., & Dejkriengkraikul, N. (2021). Palliative Performance Scale and survival in patients with cancer and non-cancer diagnoses needing a palliative care consultation: a retrospective cohort study. BMC Palliative Care, 20(1), 74. https://doi.org/10.1186/s12904-021-00773-8
  • Torbahn, G., et al. (2020). Mini Nutritional Assessment (MNA®) predicts mortality, treatment outcomes and quality of life in cancer patients: A systematic review. Nutritional Reviews, 78(8), 676–691. https://doi.org/10.1093/nutrit/nuaa016
Comments on the Quality of English Language

The manuscript is written in generally understandable English and uses appropriate scientific terminology. However, certain sentences are overly long, particularly in the Methods and Results sections, which affects clarity. The narrative would benefit from professional language editing to streamline syntax, improve readability, and ensure terminological consistency.

Author Response

Replies to the reviewers

We thank all reviewers for their comments and suggestions. Their suggestions improved the quality of the manuscript.

Please find our replies below.

Reviewer 2:

1- Title and Abstract

The title is accurate and clearly reflects the central objective of the study. However, although the abstract outlines the main objectives, I suggest restructuring it according to the following criteria (https://www.mdpi.com/journal/jcm/instructions):

- Background/Objectives (state the purpose, research questions, and context),

- Methods (describe the study design, main variables, participant characteristics, and whether the study was registered, keeping in mind that clinical trials must include the elements required by CONSORT),

- Results (report the main results with key statistics),

- Conclusions (summarize the main interpretations and implications for clinical practice). I suggest this because, given the empirical nature of this study, I believe the structured format facilitates the reader's understanding.

Answer: We appreciate the reviewer's insightful comment and helpful suggestion to restructure the abstract. We have now revised the abstract according to the requested format (Background/Objectives, Methods, Results, Conclusions) to enhance clarity and improve the reader's understanding. We believe the structured format now provides a more concise and comprehensive overview of the study's purpose, methods, key findings, and implications. Thank you for helping us improve the manuscript.

2- Introduction

The introduction correctly defines Anorexia-Cachexia Syndrome as a complex, multifactorial, and highly prevalent condition in patients with advanced chronic diseases. The authors provide updated definitions and emphasize their prognostic relevance, citing landmark studies. This approach is appropriate and demonstrates familiarity with the international scientific debate.

However, the text offers room for improvement in terms of clarity of objectives and the transition from a general framework to a specific focus. The stated objective of "identifying patterns" appears overly general: it is unclear whether it refers to clinical, nutritional, biochemical, therapeutic patterns, or a combination thereof. It would be helpful to clarify this definition from the introduction, so as to allow the reader to fully understand the scope and direction of the analysis.

Answer: We appreciate the reviewer's feedback regarding the clarity of our objectives in the introduction. We agree that the original phrasing, 'identifying patterns,' was overly general and lacked sufficient specificity.

To address this, we have revised the introduction to more clearly define the scope and direction of our analysis. We have reworded the objective to focus on “analyzing patterns of nutritional decline, symptom burden, and treatment approaches related to ACS”.

The cited literature adequately covers the pathophysiological aspects of ACS, but the rationale for focusing on the home setting is not fully developed. In this area, the available literature is fragmented and often limited to hospital-based oncology settings. Further clarifying the paucity of studies in the community-based setting and briefly comparing the operational differences compared to inpatient care would strengthen the study's justification.

Answer: We thank the reviewer for pointing out the need to strengthen our justification for focusing on the home setting. We have now expanded the introduction to:

  • Explicitly highlight the limited research on ACS in community-based palliative care settings, contrasting it with the larger body of literature focused on hospital-based oncology populations.
  • Briefly compare and contrast the operational differences between community and inpatient palliative care, emphasizing the unique challenges and opportunities presented by the home setting (e.g., resource constraints, caregiver involvement, home environment).
  • State the ACS management is critical to improve quality of life.

Furthermore, the introduction of this approach could benefit from greater attention to the organizational and sociocultural aspects that influence home management of ACS: caregiver training, available resources, multidisciplinary coordination, and logistical barriers. These factors are crucial to the effectiveness of interventions and could also be discussed in light of similar experiences in other clinical areas. For example, the literature on pain management in palliative care has shown that the use of integrated scales (Mercadante, S., Lo Cascio, A., Adile, C., Ferrera, P., & Casuccio, A. (2023). Maddalena Opioid Switching Score in patients with cancer pain. Pain, 164(1), 91–97. https://doi.org/10.1097/j.pain.0000000000002669) improves the appropriateness and timeliness of therapeutic decisions. The parallel is pertinent: also for ACS, a structured and standardized approach could optimize management, especially when dealing with multiple and interconnected symptoms. Another aspect that could be developed concerns “minor” or underestimated symptoms, such as xerostomia (Murphy Dourieu, E., Lisiecka, D., Evans, W., & Sheahan, P. (2025). Xerostomia: a silent burden for people receiving palliative care - a qualitative descriptive study. BMC palliative care, 24(1), 1. https://doi.org/10.1186/s12904-024-01617-x) and dysgeusia, which the study treats in an innovative way (Ferrara, G., Palomares, S. M., Anastasi, G., Parozzi, M., Godino, L., Gazineo, D., Sguanci, M., & Mancin, S. (2025). Anosmia, dysgeusia and malnutrition in chronic kidney disease: A systematic review. Nefrologia, 45(2), 106–115. https://doi.org/10.1016/j.nefroe.2025.02.002). Including the hypothesis that proactive management of these symptoms can influence the clinical trajectory right from the introduction would provide an interpretative anchor to the results and strengthen the coherence of the argument.

Answer: We thank the reviewer for their insightful and constructive feedback on the introduction. We have now significantly revised this section to:

  • Expand on the organizational and sociocultural aspects influencing home management of ACS, including caregiver training, available resources, multidisciplinary coordination, and logistical barriers.
  • Draw parallels to other clinical areas, specifically pain management, highlighting the benefits of structured and standardized approaches (referencing Mercadante et al.). We now make the hypothesis that it will improve effectiveness.
  • Emphasize the often underestimated impact of "minor" symptoms like xerostomia and dysgeusia and argue for their proactive management, referencing Murphy Dourieu et al. and Ferrara et al.. We now state the importance to have a closer look at this symptoms.

3- Materials and Methods

The description of the study design is clear, but the specification "without intervention" is superfluous, as the retrospective observational nature of the study already implies the absence of experimental interventions. Eliminating this term would make the text more fluent and direct.

Answer: We appreciate the reviewer's suggestion regarding the redundancy of the phrase "without intervention." We have removed this term from the description of the study design for greater clarity and conciseness.

The section on inclusion and exclusion criteria is well-structured, but raises a relevant question: among the diagnoses considered, chronic kidney disease (CKD), a condition often associated with cachexia in palliative care, is not included. It would be appropriate to specify whether CKD was absent from the cohort, undetected, or deliberately excluded, and to justify the choice, as its inclusion could have enriched the comparative analysis.

Answer: We thank the reviewer for raising this important point regarding the absence of chronic kidney disease (CKD) in our cohort. Unfortunately, due to the fact this was a retrospective study, all patients with CKD were excluded, primarily due to incomplete data sets. In prospective study, we will be sure to be more inclusive of patients, so that we can enrich the comparative analysis regarding the CKD condition, and look for factors that impacts the Anorexia Cachexia Condition. We added and highlighted a sentence about this absence in results section (lines 216-218).

The description of the collected variables (lines 104–113) is exhaustive, but the presentation in a single, very long sentence compromises readability. Grouping the data into thematic subcategories—sociodemographic, clinical, functional, symptomatic, and therapeutic—would enhance clarity and allow the reader to better navigate the flow of information.

Answer: We express our sincere appreciation for the insightful comments provided by the reviewers. In light of these and other suggestions, we have striven to thoroughly address every opinion and suggestion and improve the quality and clarity of this manuscript..

The list of statistical tests used (lines 123–134) is also comprehensive, but there is a lack of correspondence between each test and the specific hypothesis or variable analyzed. Explicitly stating, for example, that the Student t-test was used to compare means of two groups in continuous variables, while the chi-square test was used to analyze categorical distributions, would improve methodological transparency and replicability.

Answer: We thank the reviewer for highlighting the need for greater methodological transparency. To address this, we have clarified the purpose of all tests used.

Looking ahead, the manuscript could benefit from a flowchart illustrating not only recruitment but also the distribution of the main variables, thus providing a quick overview of the dataset.

Answer: Thank you for the suggestion. We believe we have addressed this point through the revised text in the methods section. We feel this provides the requested overview of the dataset and variable distribution.

4- Results

The results are rich and detailed, but contain redundancies that make them difficult to read. The correlation between baseline anorexia and baseline fatigue (r = 0.459; p = 0.001) is reported twice (lines 155 and 166), as is the association between dysphagia and xerostomia (lines 191 and 202–203), and the correlation between initial MNA and MNA at one week (lines 236–237 and 268–269). Eliminating repetition would make the section more streamlined.

Answer: Thank you for pointing out the redundancies in the results section. We have carefully reviewed the manuscript and removed the repeated information to create a more streamlined and concise presentation of the findings.

The tables are generally clear, but some include overlapping parameters that could be merged. The text is predominantly descriptive; adding brief interpretative comments immediately following key associations would help maintain the reader's attention and connect the data to clinical practice. Of particular note is the counterintuitive finding that patients with NGT/PEG exhibit greater weight loss than those without enteral nutrition. This aspect, also central to the literature (Holm et al., 2007), should be highlighted more prominently in the results, anticipating the discussion of possible pathophysiological and clinical explanations and better aligned with recent international literature.

Answer: Thank you for your helpful feedback on the presentation and interpretation of the results. We agree that merging some tables and adding interpretive comments will improve clarity and engagement. We attempted to merge the tables as suggested, and were successful in combining Tables 5 and 6. However, due to significant differences in the statistical methods used to generate the other tables, a meaningful and accurate combination was not possible.

However, we think that the Results section should primarily present findings without extensive interpretation. We agree that a detailed discussion of the NGT/PEG results and their potential explanations is best reserved for the Discussion section, where it can be properly contextualized within the existing literature.

In the Discussion section, we elaborated on the counterintuitive NGT/PEG finding, incorporating relevant literature like Holm et al. (2007) to explore the pathophysiological and clinical factors that may contribute to this phenomenon.

5- Discussion
The manuscript's discussion is undoubtedly clinically relevant, as it addresses Anorexia-Cachexia Syndrome (ACS) in a little-explored home setting. The authors highlight the syndrome's transversal nature with respect to diagnoses, the central role of oral symptoms, and treatment pathways that often begin too late. However, to enhance the text's scientific validity, I suggest enriching it with references to recent literature.

The finding that patients receiving enteral nutrition (NGT/PEG) unexpectedly experience greater weight loss deserves in-depth analysis. The literature argues that in patients with advanced cachexia—particularly those with dementia—enteral nutrition neither improves survival nor halts nutritional decline, especially if initiated at an already refractory stage. This may be due to the selection of the most compromised patients or to reverse causality: the decision to administer artificial nutrition is often made when weight loss is already evident. Guidelines and prospective studies indicate that support modalities must be balanced, individualized, and proportionate to the patient's trajectory (Amano et al., 2021; ASCO, 2020).

Another distinguishing aspect of the study is its focus on symptoms that tend to be underestimated in clinical practice, such as dry mouth, dysgeusia, and dysphagia. Several studies, particularly in head and neck oncology, have documented their substantial impact on reduced food intake, nutritional compromise, and deterioration in quality of life (Bressan et al., 2016; Ghias, 2023). Including the importance of proactive management of these symptoms in the discussion and suggesting multimodal therapeutic approaches—combining pharmacological, nutritional, and speech therapy support—would enhance the originality of the research.

The concept of a multimodal approach to cachexia, including nutrition, physical activity, and inflammation management, is now well-established (Arends et al., 2021), and is the basis of recent clinical trials such as MENAC (Multimodal Exercise, Nutrition, and Anti-inflammatory medication for Cachexia), which show a slower rate of weight loss compared to standard care. Your study implicitly suggests this direction: a direct reference to this evidence would strengthen the implicit proposal for a synergistic care plan.

The analysis of results related to functional outcomes, measured through the Palliative Performance Scale (PPS), can be further strengthened by citing studies that confirm its prognostic validity. For example, Prompantakorn et al. (2021) demonstrated that the PPS is a reliable indicator of survival in both cancer and non-cancer patients, providing essential support for advance care planning and resource allocation in home settings.

Similarly, the importance of MNA as a prognostic tool should be highlighted. A systematic review has shown that low MNA correlates with mortality, disease progression, and worsening quality of life (Torbahn et al., 2020). The integration of PPS and MNA emerges as a valid strategy for identifying frail patients who could benefit from personalized home support pathways.

Finally, it should be noted that late recruitment into the palliative care network—also observed in the present study—is consistent with evidence showing that earlier referrals are associated with less aggressive care, greater alignment with patient wishes, and in many cases improved quality of life (Temel et al., 2014; Mukamel et al., 2023). Including this link would contribute to a more robust organizational framework.

  • Amano, K., et al. (2021). Effects of enteral nutrition and parenteral nutrition on survival in patients with advanced cancer cachexia: Analysis of a multicenter prospective cohort study. Clinical Nutrition, 40(3), 1168–1175. https://doi.org/10.1016/j.clnu.2020.07.027
  • Arends, J., et al. (2021). Cancer cachexia in adult patients: ESMO Clinical Practice Guidelines. Annals of Oncology, 32(12), 1231–1245. https://doi.org/10.1016/j.annonc.2021.12.004
  • ASCO (2020). Roeland, E. J., et al. Management of cancer cachexia: ASCO Guideline. Journal of Clinical Oncology, 38(20), 2432–2452. https://doi.org/10.1200/JCO.20.00611
  • Bressan, V., et al. (2016). The effects of swallowing disorders, dysgeusia, oral mucositis and xerostomia on nutritional status, oral intake and weight loss in head and neck cancer patients: A systematic review. Cancer Treatment Reviews, 45, 105–119. https://doi.org/10.1016/j.ctrv.2016.03.009
  • Ghias, K. (2023). The impact of treatment-induced dysgeusia on oral intake and nutritional status. Cancer Reports, e1792. https://doi.org/10.1002/cnr2.1792
  • Prompantakorn, P., Angkurawaranon, C., Pinyopornpanish, K., Chutarattanakul, L., Aramrat, C., Pateekhum, C., & Dejkriengkraikul, N. (2021). Palliative Performance Scale and survival in patients with cancer and non-cancer diagnoses needing a palliative care consultation: a retrospective cohort study. BMC Palliative Care, 20(1), 74. https://doi.org/10.1186/s12904-021-00773-8
  • Torbahn, G., et al. (2020). Mini Nutritional Assessment (MNA®) predicts mortality, treatment outcomes and quality of life in cancer patients: A systematic review. Nutritional Reviews, 78(8), 676–691. https://doi.org/10.1093/nutrit/nuaa016

Answer: Thank you for these insightful comments and suggestions for strengthening the Discussion. We particularly appreciate the emphasis on the importance of referencing recent literature and expanding on the implications of our findings. I agree that incorporating the concepts of individualized, multimodal approaches to ACS management, along with evidence supporting the prognostic validity of PPS and MNA, will significantly enhance the scientific rigor and clinical relevance of the manuscript. We will address each of your points in the revised version.

Reviewer 3 Report

Comments and Suggestions for Authors

The manuscript presents results in a clear and logically organized manner, supported by well-structured tables and statistical data. The discussion is grounded in current and classical literature, with coherent interpretation of findings. The integration of nutritional, functional, and symptom-related variables is particularly valuable, and the breadth of references enhances the paper’s academic credibility.

While the study is thorough, certain sections would benefit from greater clarity and conciseness. Several passages—especially in the results and discussion—repeat data already shown in tables without adding interpretive value. Streamlining these sections and focusing on the clinical and scientific implications would improve readability.

Terminology should be standardized throughout, particularly for “ACS/SAC” and “NGT/PEG.” Table formatting could also be more consistent, ensuring uniform highlighting of statistically significant results and avoiding isolated terms (e.g., “p”) in separate lines.

Minor typographical and formatting issues should be corrected (e.g., “Hearth failure” → “Heart failure”), along with adjustments to punctuation and sentence connectors to improve flow.

Some results are presented without exploring their physiological basis or practical relevance—for instance, the association between xerostomia and PPS could be linked to prognostic and early intervention strategies. Non-significant associations could be described more briefly to keep the focus on key findings.

The limitations section is solid but could expand on potential confounding factors (e.g., the link between dexamethasone use and patient severity) and the possible bias in recording subjective symptoms such as fatigue or anorexia.

Finally, the conclusion could be more succinct, ending with a focused statement on practical implications—how the findings might guide clinical protocols or risk assessment in home palliative care.

Author Response

Replies to the reviewers

We thank all reviewers for their comments and suggestions. Their suggestions improved the quality of the manuscript.

Please find our replies below.

Reviewer 3:

While the study is thorough, certain sections would benefit from greater clarity and conciseness. Several passages—especially in the results and discussion—repeat data already shown in tables without adding interpretive value. Streamlining these sections and focusing on the clinical and scientific implications would improve readability.

Answer: We appreciate the reviewer's feedback regarding clarity and conciseness. We have carefully reviewed the manuscript, particularly the Results and Discussion sections, and have streamlined several passages by removing redundant data and focusing on the clinical and scientific implications of our findings. We believe that these revisions have significantly improved the readability and overall impact of the manuscript.

Terminology should be standardized throughout, particularly for “ACS/SAC” and “NGT/PEG.” Table formatting could also be more consistent, ensuring uniform highlighting of statistically significant results and avoiding isolated terms (e.g., “p”) in separate lines.

Answer: Thank you for pointing out the inconsistencies in terminology and table formatting. We have carefully reviewed the manuscript to standardize the use of "ACS" and "NGT/PEG" throughout. Additionally, we have revised the table formatting to ensure consistent highlighting of statistically significant results and to eliminate any isolated terms or formatting issues. We have made every effort to address these points and hope that the revised manuscript meets your expectations.

Minor typographical and formatting issues should be corrected (e.g., “Hearth failure” → “Heart failure”), along with adjustments to punctuation and sentence connectors to improve flow.

Answer: Thank you for identifying the minor typographical and formatting issues. We have carefully reviewed and corrected these throughout the manuscript, including changes like "Hearth failure" to "Heart failure," as well as adjusting punctuation and sentence connectors to improve the overall flow. We appreciate your attention to detail.

Some results are presented without exploring their physiological basis or practical relevance—for instance, the association between xerostomia and PPS could be linked to prognostic and early intervention strategies. Non-significant associations could be described more briefly to keep the focus on key findings.

Answer: Thank you for this suggestion. We've revisited the results presentation, expanding on the physiological basis and practical relevance of key findings, such as the association between xerostomia and PPS, which we've now linked to prognostic and early intervention strategies in the Discussion. We've also streamlined the reporting of non-significant associations to maintain focus on key findings. Integrating your comments, along with those from the other reviewers, has, we believe, resulted in a significantly strengthened Discussion and more impactful Conclusions.

The limitations section is solid but could expand on potential confounding factors (e.g., the link between dexamethasone use and patient severity) and the possible bias in recording subjective symptoms such as fatigue or anorexia.

Answer: Thank you for highlighting the limitations section. We agree that expanding on potential confounding factors and biases related to subjective symptom reporting will strengthen the manuscript. We will incorporate these points into the revised limitations section.

Finally, the conclusion could be more succinct, ending with a focused statement on practical implications—how the findings might guide clinical protocols or risk assessment in home palliative care.

Answer: Thank you for your suggestion to make the conclusion more succinct and focused on practical implications. We have revised the conclusion to address this point.